# Investigation of Solid Phase Microextraction Gas Chromatography–Mass Spectrometry, Fourier Transform Infrared Spectroscopy and $^1$H qNMR Spectroscopy as Potential Methods for the Authentication of Baijiu Spirits

**Neil Fitzgerald [1,\*]** and **John C. Edwards [2]**

[1] Department of Chemistry, Biochemistry and Physics, Marist College, 3399 North Road, Poughkeepsie, NY 12601, USA

[2] Process NMR Associates, LLC, 84 Patrick Lane, Ste 115, Poughkeepsie, NY 12603, USA

\* Correspondence: neil.fitzgerald@marist.edu; Tel.: +1-845-575-3000

**Abstract:** The baijiu spirit is often the focus of fraudulent activity due to the widely varying prices of the products. In this work, Solid Phase Microextraction Gas Chromatography (SPME GCMS), Fourier Transform Infrared (FTIR) Spectroscopy and $^1$H qNMR spectroscopy were evaluated as potential methods to authenticate baijiu samples. Data were collected for 30 baijiu samples produced by seven different distilleries. The data from the SPME GCMS and FTIR methods were treated by a Principal Component Analysis to identify clusters that would suggest chemical differences in the products from different distilleries. The results suggest that SPME GCMS has the potential to be a fully portable method for baijiu authentication. FTIR did not appear suitable for authentication but can be used to find the %ABV range of the sample. $^1$H quantitative NMR ($^1$H qNMR) was utilized to quantify the ethanol concentrations and calculate the observable congener chemistry comprising ester, ethanol, methanol, fusel alcohol, and organic acids. Discrepancies in ethanol content were observed in three samples, and a lack of major congeners in two samples indicates the possible presence of a counterfeit product. Detailed and quantitative congener chemistry is obtainable by NMR and provides a possible fingerprint analysis for the authentication and quality control of baijiu style, producer, and the length of the ageing process.

**Keywords:** baijiu; FTIR; $^1$H NMR; qNMR; SPME GCMS; PCA; authentication; PLS; furfural; esters; acids; fusel; chemometrics





## 1. Introduction

Baijiu is considered to be the "national liquor of China" and, with roughly 17 billion liters produced each year, it is the world's best-selling liquor [1]. The beverage is produced from red sorghum, water, and other grains. Typically, the grains are mixed and steamed prior to the addition of a fermentation agent. The fermented product is then distilled and matured in earthenware pots. Baijius can be categorized based on flavor and aroma characteristics. Examples include strong flavor, light flavor, sauce flavor, rice flavor, feng flavor, laobaigan flavor, te flavor, sesame flavor, jiugui flavor, herbal flavor, yubingshaojiu flavor, and mixed flavor. Baijiu flavor and aroma also vary by the maturation time and distillery. With some bottles of baijiu costing hundreds to thousands of dollars or more (in 2021, 24 bottles of vintage baijiu were sold at auction for USD 1.4M), the counterfeit industry is lucrative and widespread [2]. The high costs are primarily due to the inability of producers to supply the increasing demand. Baijiu is particularly vulnerable to fraud due to numerous technical opportunities, strong economic drivers, and a lack of adequate control measures [3]. Some producers have sought to reassure consumers that they are purchasing a genuine product by using increasingly complex packaging and anti-piracy

techniques on their bottles. Counterfeiters, however, have responded to this and may reuse empty genuine bottles and packaging. Methods to determine the authenticity of the liquid are, therefore, sought after.

Recently, Burns et al. [4] demonstrated that fluorescence spectroscopy, combined with a suitable statistical method, can be used to authenticate baijiu spirit samples. While a fast and inexpensive method, the instrument used is confined to a laboratory setting and does not provide any information about the chemical differences in the samples. Zhang et al. [5] identified the origins of Chinese baijiu samples using a combination of organic acids, trace elements, and stable carbon isotope ratios. These data were combined and analyzed by a Principal Component Analysis (PCA) and Partial Least-Squares Discriminant Analysis (PLS DA). Zhao et al. [6] used gas chromatography to characterize key aroma compounds in Chinese baijiu. They highlight the importance of aroma differences in baijiu samples. In recent years, Solid Phase Microextraction (SPME) has become a common method to determine aroma compounds in beverages when combined with Gas Chromatography–Mass Spectrometry (GCMS). For example, SPME GCMS has been used to identify aroma compounds in beer [7], wine [8], and whiskey [9]. The SPME sampling technique has the advantages of being rapid, simple, and solvent free. It can also identify compounds present in the aromas. Moreover, it can be combined with portable GC-MS to produce a fully portable method.

Principal Component Analysis (PCA) is a method for reducing the dimensionality contained in a large data set in order to identify patterns. PCA is an unsupervised method that allows for data with a large number of features to be viewed in two or three dimensions as a scores plot. Samples with similarity will tend to cluster together on the plot. The loadings can then be investigated to determine the sources of differences and similarities. For example, using PCA, it was possible to classify whiskey using gas chromatography data [10]. Wang et al. [11] employed PCA as a tool to classify baijiu styles based on non-volatile organic acid content detected by HPLC and LC-MS after derivatization. Burns et al. [4] used the similar chemometric technique of Partial Least-Squares-Discriminant Analysis (PLS-DA) to authenticate baijiu samples based on fluorescence spectra.

[1]H qNMR spectroscopy has developed into an excellent tool for food analysis and authentication because all proton chemistry is observed in a quantitative and inherently linearity manner [12–18]. Though sensitivity to minor components is not a strength of the technique (sub 5–50 mg/L depending on the molecule), the major components of a food or beverage can be observed with little sample preparation, and a single experiment observes multiple component types at the same time. Organic acids, carbohydrates, amino acids, alcohols, esters, aldehydes, lipids, botanicals, terpenes, etc., can all be observed in the context of the entirety of the sample. This means that the technique identifies the components that are expected to be observed but also readily identifies unexpected components. This allows fingerprint and database approaches to be developed for food and beverage markets to authenticate products and identify fraud. [1]H metabolomics and quantitative qNMR approaches can be used to screen for authenticity and quality control at any step in the production, transportation, and sales process [19–24].

In this work, we investigate portable SPME-GCMS, Attenuated Total Reflectance (ATR) Fourier Transform Infrared (FTIR) Spectroscopy, and quantitative Nuclear Magnetic Resonance (qNMR) as potential methods for authenticating baijiu samples. FTIR was considered, as it is a common analytical instrument and is capable of providing data quickly with little or no sample preparation. It can also provide information on the chemicals present in the sample. FTIR data have been used previously to classify vegetable oils, fruit juices, and coffee when combined with a Principal Component Analysis (PCA) [25–28]. SPME combined with portable GCMS was investigated, as it known to be capable of classifying beverages based on aroma compounds and has the potential to be used to test suspect samples onsite.

## 2. Materials and Methods

### 2.1. Baijiu Samples

Thirty unique samples of baijiu, obtained from seven different distilleries, were provided by Brewing and Distilling Analytical Services (Lexington, Kentucky). The samples are listed in Table 1.

**Table 1.** Samples correlated to distillery and country of origin.

| Distillery Number | Number of Samples | Location of Distillery |
|---|---|---|
| 0 | 21 | Taiwan |
| 1 | 3 | China |
| 2 | 1 | China |
| 3 | 2 | China |
| 4 | 1 | China |
| 5 | 1 | USA |
| 6 | 1 | China |

### 2.2. SPME-GCMS

All thirty samples were screened using a portable GCMS (Perkin Elmer, Torion T9) with an MXT-5 column. This instrument can be fitted with a small carrier gas canister and battery pack, in order to be a fully portable instrument. For identification, a vial containing the sample was opened to the atmosphere and the headspace exposed to a Custodion Solid Phase Micro Extraction fiber (DVB/PDMS, 65 μm) for 30 s prior to injection into a Torion T-9 GCMS (MXT-5 column, 50–296 °C at 2 °C/s with an initial hold time of 10 s and final hold time of 47 s). This sampling method was chosen for simplicity, as it would allow the analysis of samples on-site without the need for transferring to sealed vials, the addition of chemicals (e.g., salts), or additional equipment (such as a heater or stir plate). Compounds were identified using the NIST MS search 2.0 program and by a comparison of retention times to compounds identified in previous samples obtained using identical conditions.

### 2.3. FTIR

For the FTIR analysis, a Nicolet iS5 FTIR-ATR spectrometer with a single bounce diamond in ATR mode was employed. Absorbance spectra were obtained between 600 cm$^{-1}$ and 4000 cm$^{-1}$, collecting 7053 points at a resolution of 1 cm$^{-1}$ and exported in SPC format. The data were processed in Spectragryph Optical Spectroscopy software version 1.2.17d (Oberstdorf, Germany). Moreover, 1.2.14. Partial Least-Squares (PLS) regression analyses were performed using Eigenvector Research Solo version 7.0.3 (Manson, WA, USA).

### 2.4. H qNMR

For the $^1$H qNMR analysis, a Varian MercuryMVX-300 spectrometer operating at 299.99 MHz with a Varian 5 mm ATB probe was utilized to produce $^1$H NMR spectra with a standard pulse sequence s2pul, with a pulse width of 10 μs, a 10 s relaxation delay, and a spectral width of 7.2 kHz collecting 64 k complex FID points, over a 8.9 s acquisition time, averaging 64 scans. Data were processed with Mestrelab MestreNova v.14.1.1-24571 (Santiago, de Compostela, Spain). The temperature during the acquisition was 300K.

NMR Materials: TMSP (added at 47.20 mg in 100 mL of $D_2O$) was used as the chemical shift reference. $D_2O$ (99.9%D) was purchased from CortecNet Corp (Brooklyn, NY, USA). TMSP-2,2,3,3-D4 (D 98%) was purchased from Cambridge Isotopes Laboratories (Andover, MA, USA). No buffer was used in the deuterated solvent. Maleic acid internal standard was purchased from Sigma-Aldrich (TraceCERT Standard for quantitative NMR—Product Number 92816, Lot BCCC6481, Exp Jan 2024, Certified Purity 99.94% +/− 0.15%).

### 2.5. qNMR Calculations

The standard solution for qNMR analyses is 10.0 mg of maleic acid (MA) per 100 mL in $D_2O$ solution.

Baijiu Sample: The $^1H$ NMR data for quantitation of the volatile components found in Baijiu were obtained on 175 µL of Baijiu, which was placed in a 5 mm tube with a 100 µL aliquot of standard maleic acid solution (equivalent to 10 mg). The calculation utilized to quantify these components was:

$$\text{Comp (mg/L)} = 0.995 \times 10 \text{ mg} \times ((\text{I}_{comp}/\text{N}_{comp})/50) \times (\text{Mw}_{comp}/116.1) \times (1,000,000/175)$$

where Comp is the calculated weight of the component in mg/L, which is derived from the weight of maleic acid (MA) internal standard (10 mg); $I_{Comp}$ = integration of component resonance; $N_{Comp}$ = number of protons integrated; $I_{MA}/N_{MA} = 50$ (MA integral set as 100); $Mw_{Comp}$ = molecular weight of component molecule; $M_W$ of MA = 116.1 amu; and the 1,000,000 mL /175 mL factor rectifies the volumetric component of the calculation to allow mg/L to be calculated. A value of 0.995 represents the 99.5% purity of the maleic acid standard.

Once the mass of ethanol per liter is known from the above qNMR calculation, the volume that this mass represents is calculated using a density of ethanol at 20 °C (0.7892 g/mL). The volume of water per liter is calculated by subtracting the volume of ethanol per liter (modified by 3%, which is a typical volume contraction for spirits in the 30–60 %ABV range). Using these values, the mole fraction of ethanol is calculated. Tables and plots are available showing the relationship between the mole fraction of ethanol and the volume contraction that occurs upon mixing ethanol and water. The ethanol content is then calculated by applying this contraction on a percentage basis to the theoretical volume calculated from the mass of ethanol in the sample.

### 2.6. Statistical Analysis

The statistical analysis of the data was performed using a free online chemometrics program MetaboAnalyst 5.0 (https://www.metaboanalyst.ca (accessed on 30 December 2022)). FTIR spectra were collected in SPC format, integrated using the Spectrogryph program (F. Menges, Version 1.2.14, 2020, http://www.effemm2.de/spectragryph (accessed on 4 June 2020)) and results combined as a CSV file in order to be introduced to the MetaboAnalyst program for statistical analysis. Data were normalized by sum and auto-scaled.

SPME-GCMS peaks were manually integrated and peak areas tabulated in MS Excel. The peak area of the first observed peak was removed from the data. This peak appears to be a mixture of highly volatile compounds (including ethanol) and could not be positively identified. Peaks were identified where possible by matching mass spectra and retention indexes to compounds in the NIST database. Data were normalized by sum and auto-scaled prior to PCA.

Partial least-squares regression models were performed using Eigenvector Research Solo Version 7.0.3 (Manson, WA, USA). The FTIR data were mean-centered and regressed against ethanol values calculated on the samples by $^1H$ qNMR.

### 3. Results

The Principal Component Analysis was performed on the FTIR and SPME-GCMS data separately. Figure 1 shows the 2D PCA scores plot based on the FTIR data for samples from distillery 0 and distillery 1. These distilleries were chosen as they had more samples. No significant difference could be discerned between distilleries. None of the other distilleries could be positively differentiated based on the FTIR data. Performing the PCA using the SPME-GCMS data, however, did show some distinct patterns. The scores plot using SPME-GCMS data, accounting for 60.4% of the data variance, is shown in Figure 2. Samples from distillery 1 are clearly different to distillery 0. The biplot showed that the major chemical differences between distillery 0 and 1 were an increase in ethyl esters (specifically ethyl propanoate, ethyl butanoate, ethyl isobutanoate, ethyl pentanoate, and ethyl hexanoate)

in samples from distillery 1 compared to distillery 0. Similar chemical differences were also true, to a lesser extent, for distillery 3, although only two samples were available from this distillery. Distilleries 4 and 6 were also found to be discernable using SPME-GCMS data. Only single samples were available from each of these distilleries. Both samples had a distinct lack of ethyl esters, containing only a few small peaks associated with alkanes and alkenes. Ethyl esters form via esterification reactions between organic acids and ethanol. Zhang et al. [28] found that long-chain fatty acid ethyl esters were absent from baijiu samples that had been matured for less than a year; however, short and medium chain fatty acid ethyl esters were detected. It is suspicious, therefore, that no ethyl esters could be detected in these samples. qNMR spectra also demonstrated a lack of ethyl esters in these two samples.

Figure 3 shows the PCA scores plot for all samples categorized by percent alcohol by volume (%ABV) range. This suggests that FTIR could be used as a quick, first step in the authentication of a suspect sample by providing an approximate %ABV. Samples with high %ABV had higher absorbances at 2970 (C-H stretch), 2900, 1985, and 1048 (C-O stretch) cm$^{-1}$. It is of note that not all samples fell in the expected areas of the scores plot in terms of alcohol content. One sample, for example, listed at an ABV of 40%, appears to the right of the region of samples below 40% ABV, suggesting that the actual ABV is significantly below 40%. qNMR determined this sample to have an alcohol content of just 29% ABV.

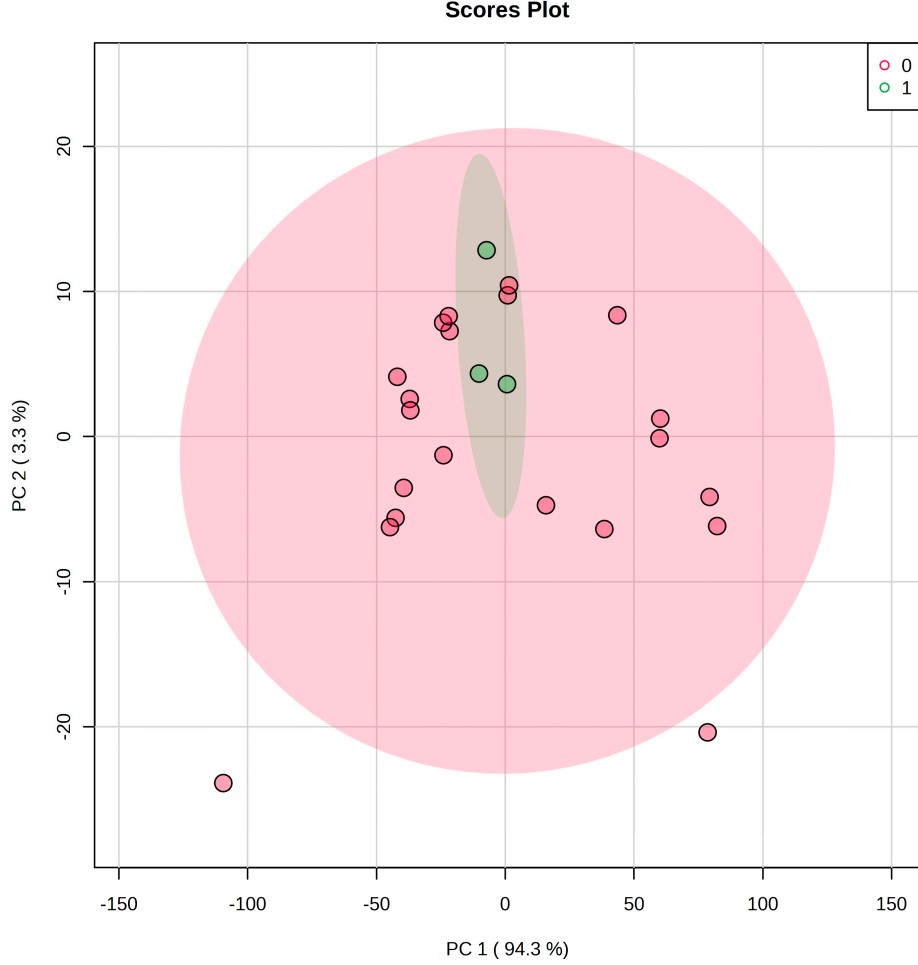

**Figure 1.** A PCA scores plot based on FTIR spectra obtained from two distilleries. Shaded areas are the 95% confidence regions.

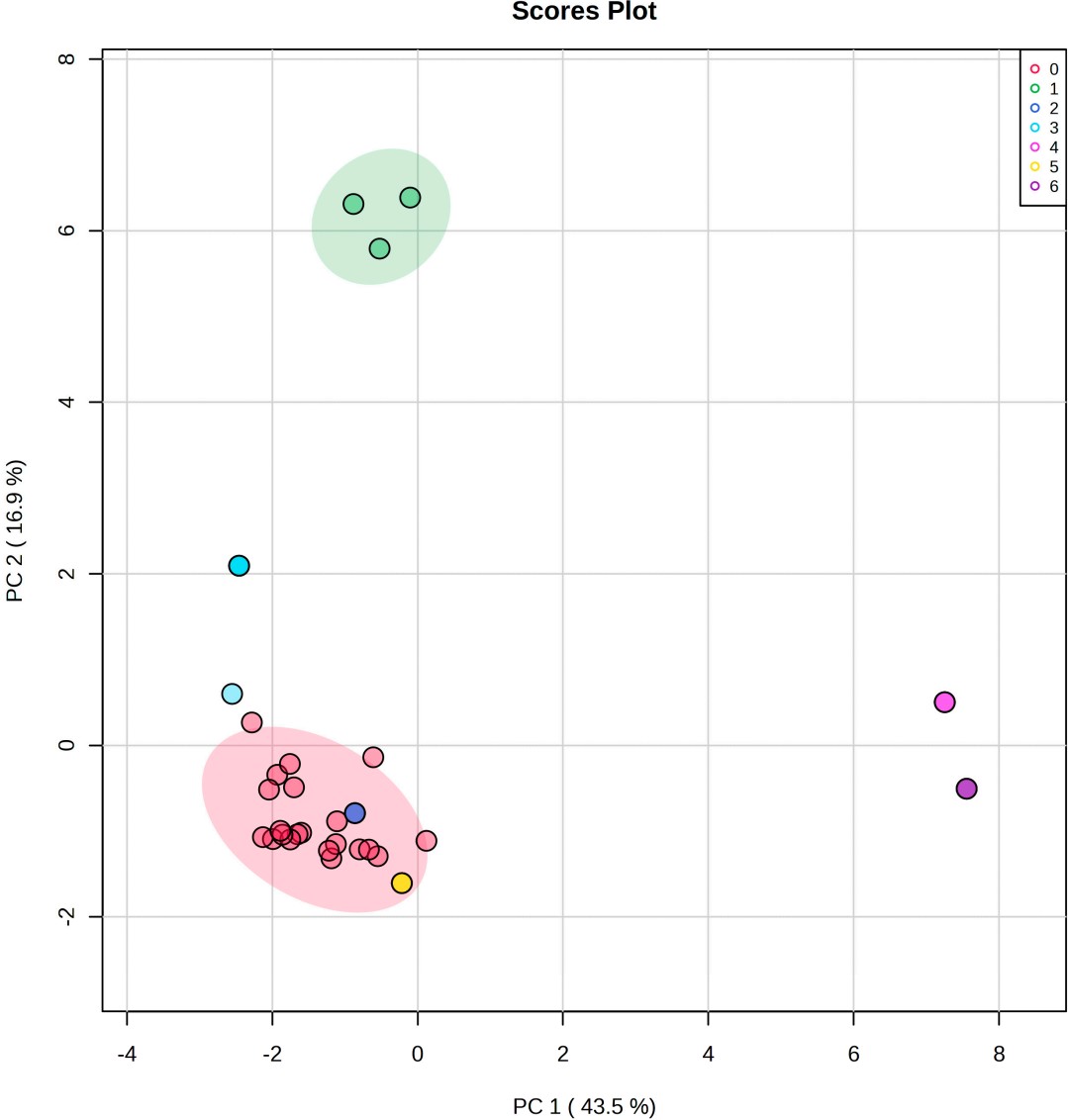

**Figure 2.** PCA scores plot of distilleries using HS-SPME-GCMS data. Shaded areas are the 95% confidence regions.

A typical baijiu [1]H-NMR spectrum is shown in Figure 4 with the assignment of the observable congeners. The spectrum is obtained using a pre-saturation pulse centered on the water peak, effectively suppressing that peak. On the instrument used to generate the NMR data, it was not possible to suppress the ethanol $CH_3$ and $CH_2$ peaks at 1.16 and 3.63 ppm. These signals are very intense and are presented on a scale such that they are greatly off-scale. The congeners that are observed are dominated by ethyl esters, which provide signals from the ethyl $CH_2$ protons in the 4.0–4.50 ppm region of the spectrum. Ethyl acetate and ethyl lactate are found to dominate most of the baijiu samples, with many of the light aroma samples showing the presence of only ethyl acetate, ethyl lactate, and lactic acid above the limit-of-detection of the experiment. Other large congener signals are observed in the 0.84–0.95 ppm region, which represent the methyl resonances of 1-propanol and isopentanol. In the 2.0–2.5 ppm region, the signals from the methyl resonances of acetaldehyde, acetic acid, and ethyl acetate can be observed. This region also shows signals from the $CH_2$ adjacent to the ester and acid carboxyls in butyric, caproate, and other ethyl esters and carboxylic acids. These higher carbon number ethyl esters are observed at much higher concentrations in the sauce aroma baijius. Furfural is also found in the sauce aroma samples along with succinic acid.

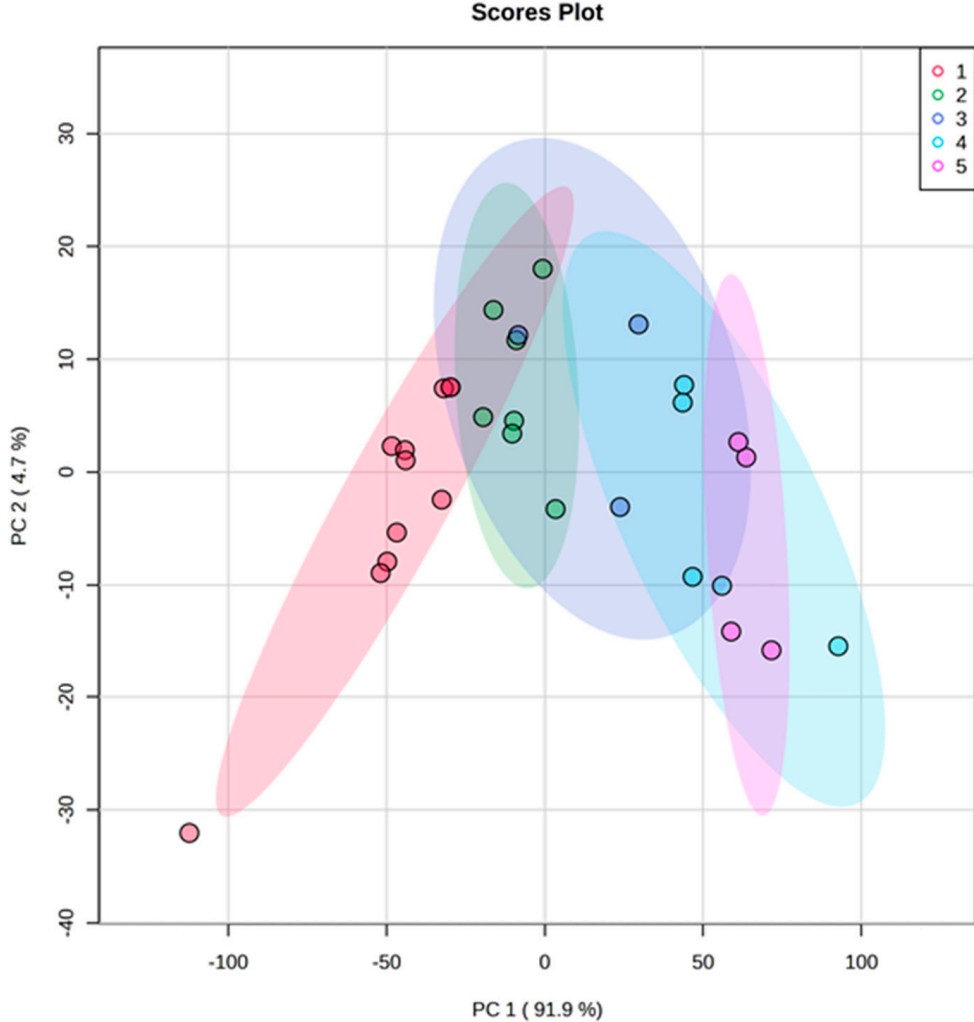

**Figure 3.** PCA scores plot of %ABV using FTIR data. Group 1—above 55%, Group 2—50–55%, Group 3—45–50%, Group 4—40–45%, Group 5—below 40%. Shaded areas are the 95% confidence regions.

Expanded plots of the major congener areas of the [1]H NMR spectrum are presented in Figures 5 and 6. In Figure 5, we have compared the methyl signals showing lactic acid, 1-propanol, isopentanol, ethyl acetate, and acetic acid. Some peak shape distortion is observed in the peaks of samples 3 and 25, which appears to be a shimming issue. Figure 6 shows the ethyl ester region. 1-propanol appears to be the principal congener alcohol produced in most of the analyzed baijiu samples. This is quite different from Western fermentations of beer, wine, and cider, which have isopentanol as the major alcohol congener. In some baijiu samples, the distribution favors isopentanol (see sample 17 for example). Three carbon congeners such as propanol, propanoic acid and 1,3-propandiol are often a sign of extensive glycerol fermentation. The sauce aroma-style baijiu samples in this study contained higher concentrations of higher carbon number esters, such as butyric and caproate, as well as showed the presence of furfural.

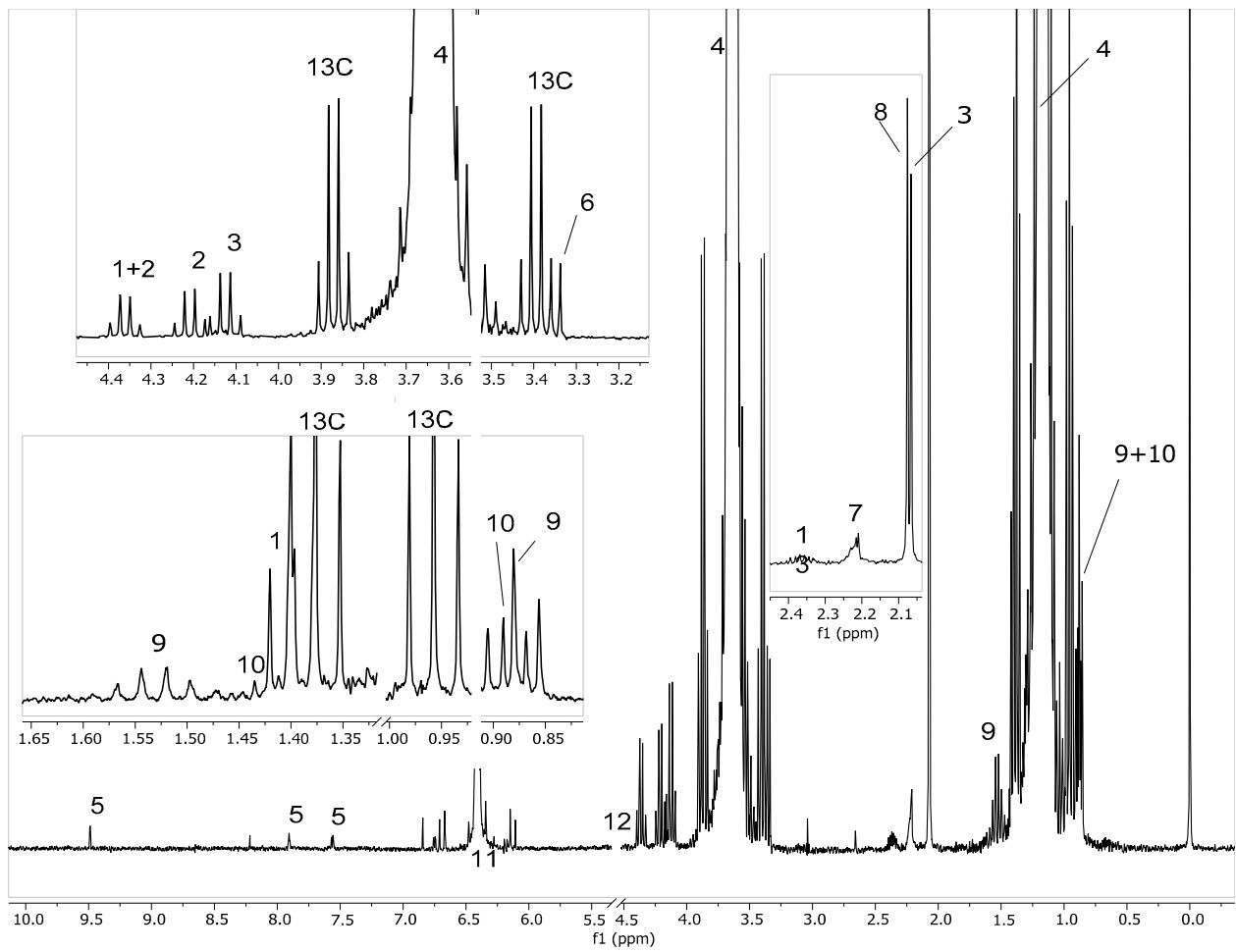

**Figure 4.** ¹H NMR spectrum of a typical baijiu sample (sample 3, manufacturer 1, Sauce Aroma) showing the assignment of various congeners that are readily observed and quantified. The numbered assignments are as follows: 1. lactic acid; 2. ethyl lactate; 3. ethyl acetate; 4. ethanol (intense signals off scale); 5. furfural; 6. methanol; 7. acetaldehyde; 8. acetic acid; 9. 1-propanol; 10. isopentanol; 11. maleic acid (internal quantitation standard); 12. water—suppressed with pre-saturation and not shown; 13. higher esters CH₂ adjacent to carboxyl—butyric, caproic etc.; 13C. ¹³C satellites arising from the coupling of intense ethanol signals with 1.1% of protons in the sample.

Utilizing the qNMR calculations described previously, the concentrations of ethanol and the major congeners were calculated for a sub-set of 10 of the baijiu samples, and these parameters are presented in Table 2. The table also contains the location and multiplicity of the peaks that were integrated to perform the calculations. It is noted that some of the baijiu samples (e.g., sample 10) do not contain observable amounts of congeners. This is often a sign of counterfeit spirits that are derived from pure ethanol [29]. Figure 7 shows a superimposed spectrum of suspected counterfeit sample 10 (green) with typical baijiu sample 3 (red). It is obvious to the observer that only ethanol is observed in sample 10, with congeners seen to be missing. Two of the thirty samples analyzed in this study showed no congeners to be present in the product. SPME-GCMS also identified the same two samples as containing very low concentrations of congeners.

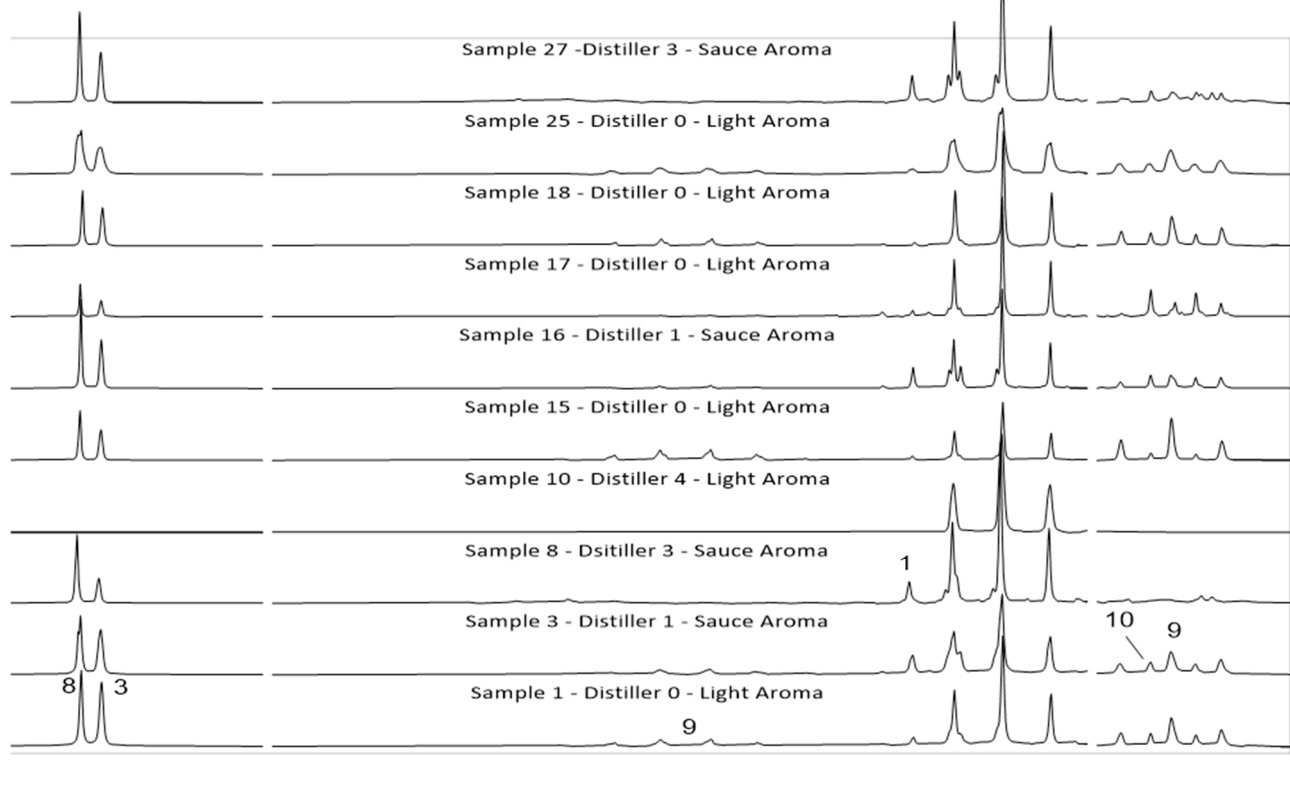

**Figure 5.** $^1$H NMR—methyl region—stacked plot comparisons of 10 of 30 baijiu samples analyzed in this study. Sample 10 stands out as a potential counterfeit sample, as it only shows the presence of ethanol. No congeners are observed, as in the other samples. The numbered assignments are as follows: 1. lactic acid, 3. ethyl acetate, 8. acetic acid, 9. 1-propanol, 10. isopentanol.

Figure 8 shows the comparison of the ethanol concentrations obtained by qNMR with the stated label values, showing an excellent correlation between the two ($R^2$ = 0.9939). With confidence in the qNMR calculation and the fact that the NMR technique directly observes and quantifies the ethanol NMR signals in the spectrum, we confirmed that 27 of the 30 baijiu samples had the correct ethanol content on the label to a standard deviation of 0.6 %ABV. However, 3 out of 30 samples had ethanol values that were considerably different to that stated on the label. Two samples from one distiller were 7% higher than stated on the label, and another distiller had a sample that was 11% lower than stated.

**Table 2.** Concentrations of ethanol and major congeners for 10 of the 30 baijiu samples analyzed.

| | | | | | Concentration (mg/L) | | | | | | |
|---|---|---|---|---|---|---|---|---|---|---|---|
| **Sample Number** | **1** | **3** | **8** | **10** | **15** | **16** | **17** | **18** | **25** | **27** | **Peaks Utilized** |
| **Aroma Style** | Light | Sauce | Sauce | Light | Light | Sauce | Light | Light | Light | Sauce | **Shift (ppm) & Multiplicity** |
| Methanol | 87 | 128 | 13 | 0 | 80 | 99 | 51 | 87 | 109 | 128 | 3.38 (s) CH3 |
| Ethanol g/L | 48.5 | 43.7 | 30.1 | 33.2 | 45.1 | 44.1 | 43.5 | 53.5 | 45.5 | 41.5 | 1.16 (t) CH3 |
| Ethanol %*v*/*v*—NMR | 59.4 | 53.5 | 37.0 | 40.7 | 55.2 | 54.0 | 53.3 | 65.7 | 55.7 | 50.8 | - |
| Ethanol %*v*/*v*—Label | 58.0 | 53.0 | 38.0 | 40.0 | 56.0 | 53.0 | 53.0 | 66.0 | 56.0 | 52.0 | - |
| Isopentanol | 385 | 486 | 0 | 0 | 431 | 454 | 700 | 417 | 448 | 240 | 0.89 (d) CH3 |
| 1-Propanol | 1017 | 1095 | 0 | 0 | 2711 | 611 | 284 | 1162 | 1010 | 393 | 0.88 (t) CH3 |
| Acetic Acid | 1107 | 1054 | 410 | 0 | 1190 | 1172 | 331 | 772 | 950 | 719 | 2.08 (s) CH3 |
| Lactic Acid | 333 | 1281 | 470 | 0 | 563 | 1405 | 9 | 27 | 346 | 771 | 3.36 (q) CH2 |
| Ethyl Acetate | 1778 | 1500 | 286 | 0 | 1760 | 1596 | 633 | 1266 | 993 | 763 | 4.12 (q) CH2 |
| Ethyl Lactate | 340 | 655 | 257 | 0 | 306 | 586 | 503 | 172 | 172 | 566 | 4.21 (q) CH2 |
| Ethyl Butyrate | 0 | 0 | 0 | 0 | 320 | 172 | 377 | 206 | 503 | 46 | 4.26 (q) CH2 |
| Ethyl Caproate | 0 | 284 | 397 | 0 | 0 | 1562 | 0 | 0 | 0 | 814 | 4.13 (q) CH2 |
| Furfural | 0 | 236 | 0 | 0 | 0 | 227 | 0 | 0 | 0 | 0 | 9.49 (d) CH |

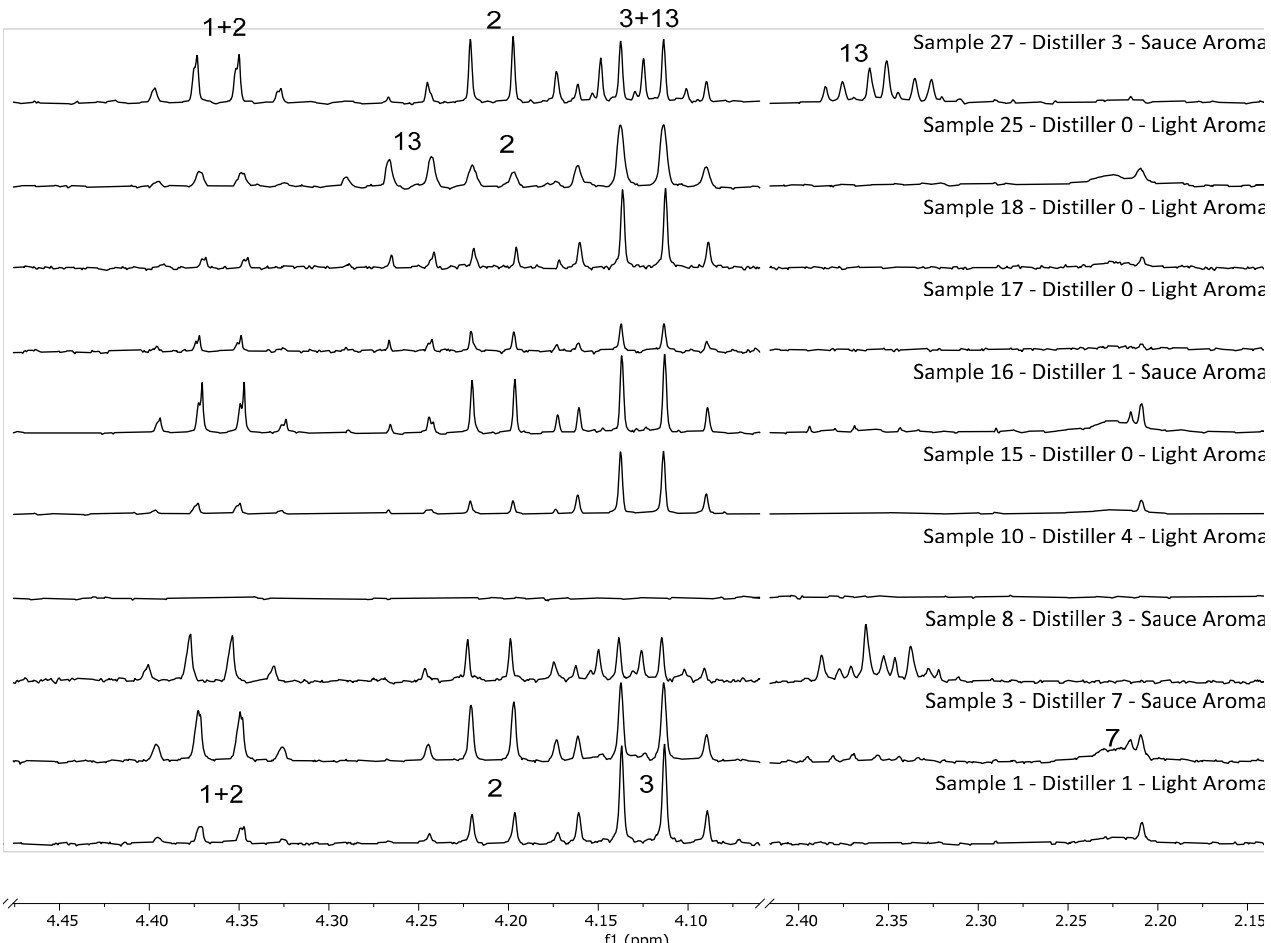

**Figure 6.** [1]H NMR—ethyl ester region—stacked plot comparisons of 10 of 30 baijiu samples analyzed in this study. Sample 10 stands out as a potential counterfeit sample, as it only shows the presence of ethanol. No congeners are observed, as in the other samples. The numbered assignments are as follows: 1. lactic acid, 2. ethyl lactate, 3. ethyl acetate, 7. acetaldehyde, 13. higher esters $CH_2$ adjacent to carboxyl—butyric, caproic.

Having confirmed the ethanol concentration of 27 of the baijiu samples by [1]H qNMR, we decided to utilize the 27-sample dataset to produce a Partial Least-Squares (PLS) regression model using the FTIR-ATR data utilized in the PCA analysis. Figure 9 shows a superposition plot of the 27 FTIR spectra obtained with an expansion showing the specific peaks at 1044 and 1086 cm$^{-1}$ that can be attributed to the C-O stretch of ethanol [30,31].

Figure 10 shows the quality of the resulting PLS regression model that correlates the [1]H qNMR ethanol content to that predicted by FTIR-ATR. The model shows an excellent correlation ($R^2 = 0.991$) and an SECV of 0.75 %ABV, which is close to the standard error associated with the NMR correlation with the label values provided on the baijiu bottles. The correlation was obtained with two latent variables and involved a cross-validation process involving a Venetian blind approach with five data splits.

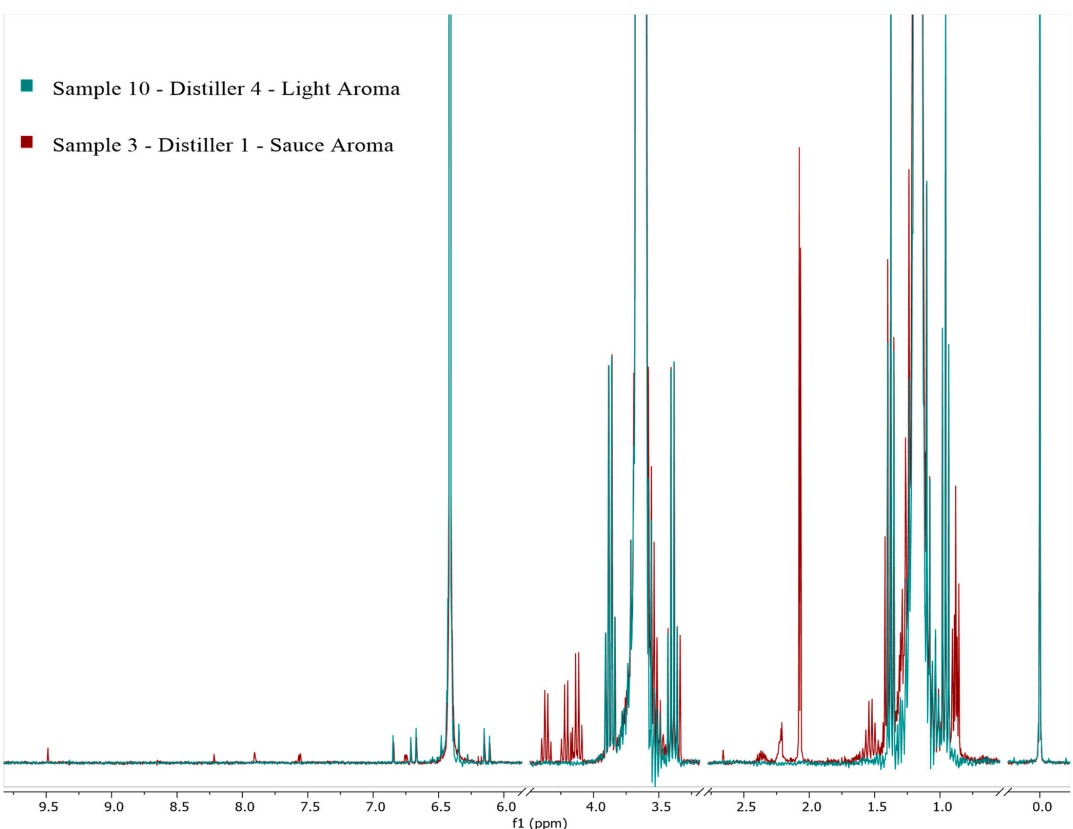

**Figure 7.** Superimposed $^1$H NMR spectra showing a suspected counterfeit baijiu (sample 10, green) containing only ethanol and no congener chemistry compared to a typical baijiu (sample 3, red), which shows the obvious congener signals from furfural (peaks 6.7 to 9.5 ppm), ethyl ester (4.0–4.5 ppm), acetaldehyde, acetic acid (2–2.5 ppm), and higher alcohols (0.85–0.95 ppm).

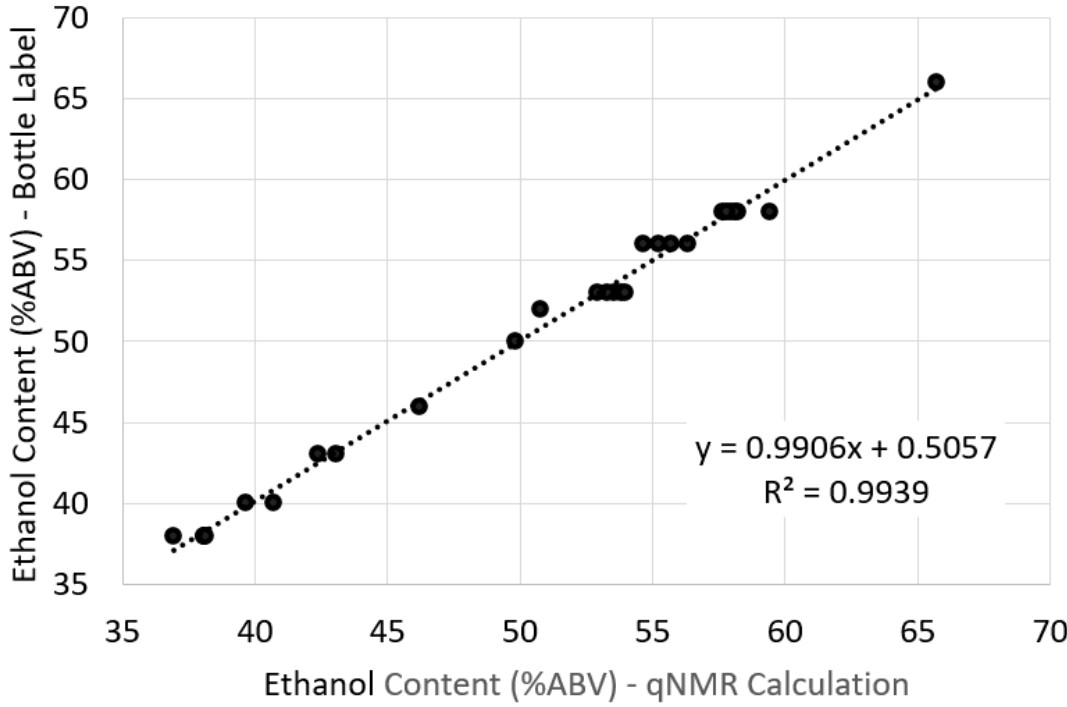

**Figure 8.** Comparison of qNMR-calculated ethanol content (%ABV) and the stated label content.

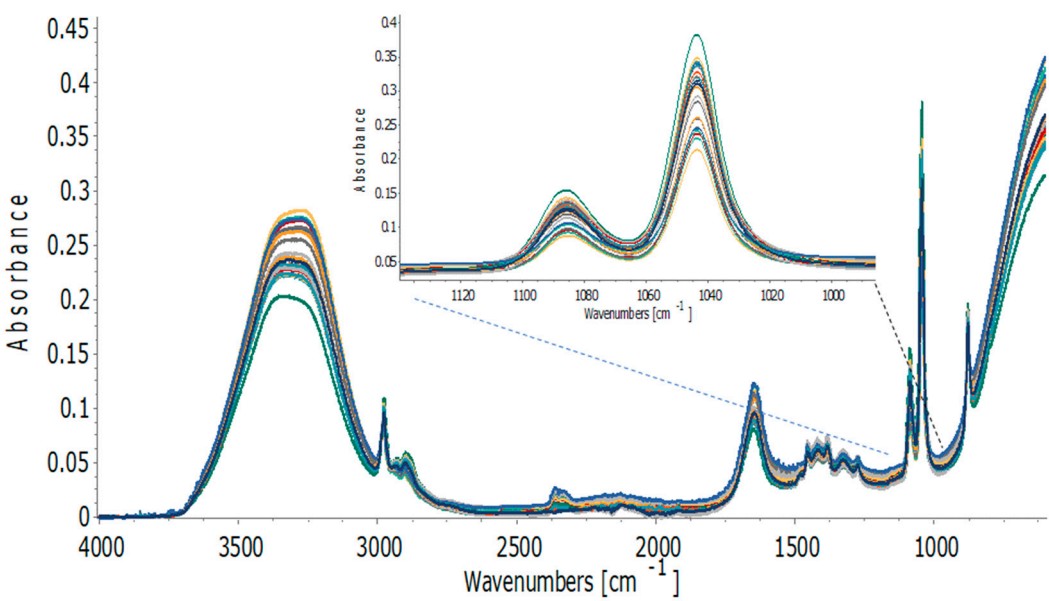

**Figure 9.** FTIR-ATR spectra obtained on 27 authenticated baijiu samples. Expansion insert shows the C-O stretch absorbances that are highly correlated to ethanol content showing the variance in these absorbances in the observed 38–66 %ABV range of the samples analyzed.

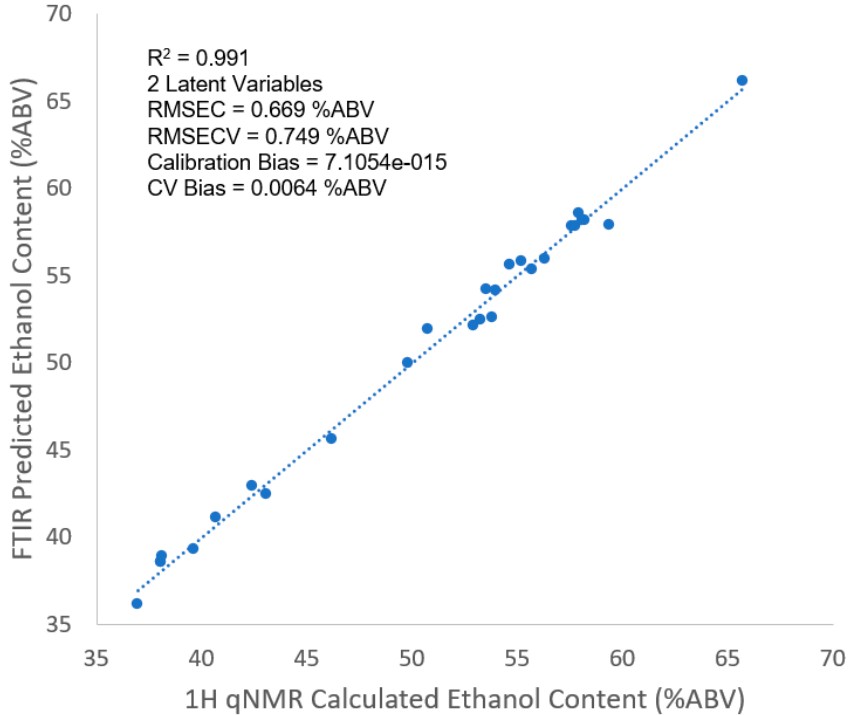

**Figure 10.** PLS correlation plot showing the prediction of ethanol content (%ABV) by FTIR-ATR spectroscopy correlated to the [1]H qNMR calculation of the ethanol content of 27 authenticated baijiu samples.

## 4. Discussion

The objective of this work was to investigate if the commonly used analytical techniques of FTIR, SPME-GCMS, and [1]H qNMR can be used to identify the source distillery of a baijiu sample for authentication purposes using an appropriate statistical treatment. The results suggest that FTIR is not an effective method to determine the origin of the baijiu samples using Principal Component Analysis (PCA). However, it does have some ability

to discern the approximate %ABV of a sample when observed directly and can be used definitively for this purpose if chemometric models are developed utilizing Partial Least-Squares regression. Excellent correlation can be observed to a standard error of prediction of 0.75 %ABV. Given the common use of FTIR and the fast, simple analysis, it could be attractive as a first screening method for a suspect sample. Further analysis of the data from 1H NMR will be applied to the exploration of using FTIR to predict the concentrations of methanol, acetic acid, and ethyl esters via PLS chemometric approaches. Once these chemometric approaches are developed, they can be used to screen for outlier samples where the FTIR prediction is very different from the reported ethanol content on the label. For example, three of the baijiu samples in this study appear to be mislabeled in terms of %ABV based on FTIR, PLS modeling outliers, and qNMR calculation. These incorrect label values may point to baijiu products that have been counterfeited or adulterated.

SPME-GCMS does appear to be able to identify sample origin in some cases. Unfortunately, multiple samples were only available for three distilleries; however, significant differences in volatile components were detected. The instrument used has the added advantages of being full portability, therefore removing the need for a physical laboratory. The analysis is fast and simple, and the identification of chemicals is also possible. For example, in this study, samples from distillery 1 clearly contained more ethyl esters, which would impart a fruitier aroma.

$^1$H NMR shows great promise to develop a rapid check of baijiu authenticity based on congener distribution. Many studies and review articles exist demonstrating the correlation between major ethyl ester and acid components with baijiu flavor and the fermentation and ageing processes applied [30,32–34]. The current NMR study demonstrates that the major esters are readily quantified, which would allow for the development of a correlation between either baijiu style, or even the production distillery and the particular ester distributions. Acid components are also observed, as their presence is necessary to form the various esters. It should be noted that the esters formed between acid components and the various fusel alcohols (1-propanol, isopentanol, isobutanol) are not observed by $^1$H NMR, as the low concentration of these fusel alcohols leads to the concentration of these minor, but sensorially important, esters to be below the limit-of-detection of the $^1$H NMR experiments performed on our instrument under the experimental conditions we can provide. Modern digital spectrometers operating at higher magnetic fields, utilizing advanced multi-peak suppression techniques performed on state-of-the-art cryogenically-cooled probes, may have an opportunity to observe these lower concentration congeners. The excellent results obtained on whisky and bourbon in recent years [21–24] demonstrate the potential for NMR to observe congeners at detection levels far lower than previously reported.

When ester and acid quantitative distribution values are also combined with the observed congener fusel alcohol distributions, the combination of these two definitive chemistries might allow for a better authentication of baijiu by style, maturity, distiller, geographical region, grain bill, etc. These same congeners can be used to identify counterfeit products that are produced from pure ethanol, thereby skipping the craft brewing and ageing processes.

Furfural was observed in some samples, which leads to questions about the limits that are imposed on its presence in many alcoholic beverages. Furfural typically develops during the fermentation stages of the process rather than developing after distillation, and it is very dependent on the brewing process, fermentation period, and temperatures. Furfural is observed at the highest concentrations in the sauce aroma style and at increasingly lower concentrations in strong and light aroma styles, respectively. It has been suggested that sensors developed to observe furfural and certain organic acids might be used in the detection of fraudulent baijiu products [35]. Plainly, $^1$H NMR analysis could be used in a similar approach to identify different styles of baijiu and from quantitation to identify the production site of the baijiu when combined with other ester, acid, and fusel quantities.

We show that a Principal Component Analysis of aroma compounds using a portable GCMS instrument is sufficient to identify suspect baijiu samples. Suspect samples can then

be investigated by qNMR. In this study, two samples were identified as possible counterfeits due to the lack of congeners observed by either SPME-GCMS or qNMR. Counterfeit or mislabeled products can also be identified from the quantitative ethanol contents calculated by the NMR. The sensitivity of the NMR technique will probably limit its application in the correlation of congener chemistry with the sensory perception of the baijiu spirits.

Oxidation products such as acetic acid and various ethyl esters can also be used to identify that a baijiu has undergone an ageing process, whereby oxidation and condensation reactions have occurred. The concentration of these components would authenticate that an aged baijiu has actually undergone an ageing process before bottling.

## 5. Conclusions

The results from this study suggest that solid phase microextraction of the vapor above baijiu liquid has potential as a quick, simple, and portable method to identify suspect baijiu samples. Potentially, the SPME GCMS equipment could be transported to a site of samples and the analyses performed by simply opening the containers and exposing the fiber to the vapor above the liquids. Nevertheless, more samples need to be evaluated to fully investigate the potential of this technique.

Fourier Transform Infrared spectroscopy does not appear to be useful for authentication purposes; however, it might be attractive to conduct a PCA of FTIR spectra on suspect liquors to quickly determine the approximate alcohol content. FTIR can certainly be used to quantify the ethanol content of baijiu and act as a screening tool to capture fraudulent samples.

$^1$H NMR spectroscopy could be developed into a rapid test for quantitative ethanol and congener analysis and allow the identification of counterfeit or mis-labeled products. Detailed studies should be carried out where quantitative congener distributions of esters, fusel alcohols, carboxylic acids, and furfural may be used to determine if an all-encompassing fingerprint analysis could be developed to authenticate baijiu by style, geographical origin, brewing process, and to capture fraudulent attempts to circulate fake or sub-standard baijiu in the market. Future work will involve the repetition of these analyses on benchtop 60–125 MHz systems that are easier than superconducting NMR systems to implement in a small laboratory environment, and which are considerably cheaper to purchase and maintain compared to superconducting systems.

**Author Contributions:** Conceptualization, N.F. and J.C.E.; Methodology, NMR, and analysis, J.C.E.; SPEM-MS and FTIR interpretation, N.F.; FT-IR data obtained by J.C.E.; PLS regression of FTIR data, J.C.E.; Resources, N.F. and J.C.E.; Data curation, both authors; Writing—original draft preparation, N.F. and J.C.E.; Writing—review and editing, both authors. All authors have read and agreed to the published version of the manuscript.

**Funding:** This research received no external funding.

**Data Availability Statement:** All data are available from the author on request.

**Acknowledgments:** The authors wish to acknowledge Gary Spedding (Brewing and Distilling Analytical Services) for providing samples and guidance with the project, and Derek Bussan (Eastern Kentucky University) for helpful discussions.

**Conflicts of Interest:** The authors declare no conflict of interest.

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
