# Peer review of "Investigation of Solid Phase Microextraction Gas Chromatography–Mass Spectrometry, Fourier Transform Infrared Spectroscopy and 1H qNMR Spectroscopy as Potential Methods for the Authentication of Baijiu Spirits"

_beverages, doi:10.3390/beverages9010025_

Round 1

Reviewer 1 Report

The topic presented by the authors of the paper is interesting because it allows, by means of well-known and generally common techniques, to carry out an analysis of the authenticity of Baijiu Spirits. The introduction of the study presents, describes and clarifies the aim of the research in an accessible way and points out a gap in the technological approach currently used. In the materials and methods section of the paper, the main criticism of the authors is the insufficient number of samples used in the analyses. Please indicate why there are such differences in the number of samples. The use of a single sample in no way confirms the results obtained for that sample. According to statistical and chemometric recommendations, a minimum of 3 samples and preferably 5 samples should be prepared. In my opinion, the results obtained for single samples (distilleries 2, 4, 5 and 6) and duplicate samples (distillery 3) can only serve as a curiosity and not as a real result. Supplementing the results with a larger number of samples should allow the results obtained to be more precise. It would also be worthwhile to focus this or future work on finding a specific compound or compounds as a kind of marker. For this purpose, I would suggest using more advanced statistical techniques in combination with artificial neural networks. On the other hand, for the moment it is mainly important to expand the number of relevant samples.

The language used in the publication is accessible to the reader however, there are grammatical errors in places. I would recommend using external help or specialised software.

Author Response

We thank reviewer 1 for the thoughtful comments. We agree that at least five samples for each distillery would be preferable and less than three does not allow for statistical methods to be effectively applied. The samples we analyzed were those submitted for taste evaluation and judging. We were not able to request additional samples. In this work, PCA of GCMS data is introduced as a means to identify suspect samples which were then investigated using NMR. We agree with the suggestion for future work.

Both authors are native English speakers. The manuscript has been reviewed for grammatical errors and any necessary corrections made.

Reviewer 2 Report

The manuscript of Fitzgerald and Edwards deals with the problem of the spirit’s authentication by several physical methods.

The authors analyzed 30 different samples of baijiu spirit coming from seven different distilleries. Solid Phase Microextraction Gas Chromatography Mass Spectrometry  (SPME GCMS) and Fourier Transform Infrared Spectroscopy (FTIR) data were analyzed using a chemometric approach (Principal Component Analysis, PCA).  PCA on SPME GCMS data allowed partial clustering of the samples according to the distillery. A major contribution to the data variance was owing to the presence of the ethyl esters. Unfortunately, the small number of samples from some distilleries prevented authors from making clear separations between all possible origins of the spirits.

FTIR data did not show significant categorization by PCA, but a direct integration of the C-O stretch absorbance allowed a linear correlation with the alcohol content of the spirit.

qNMR analysis confirmed that most chemical variance between the samples is due to the presence of the different ethyl esters.

The work may be of potential interest to the readership of Beverages, but several important issues must be taken into account before the paper is ready to be accepted.

1.     For some reason, the authors only used PCA for GSMS and FTIR data. 1H NMR data are also highly amenable to this kind of analysis that can provide more insights into the origin of the chemical diversity of the samples.

2.     While the authors stated that the used NMR spectrometer was not suitable for the simultaneous suppression of several signals of the solvents, a simple multi-point presaturation procedure should provide a very clean suppression of signals of ethanol and water simultaneously. This experiment would greatly improve sensitivity to small amounts of congeners.

3.     Some information on the experimental conditions of NMR spectra registering is lacking: the temperature, the standard used for chemical shift assignments (was is DSS?), and the processing parameters (those would be especially important if PCA analysis is undertaken).

4.     Line 133 -  the volumes should be in uL (microliters), not mL.

5.     More information is necessary on statistical analysis. Were only integral intensities of the signals used in the analysis, of the spectra were split into some bins as it is often done? 

6.     There is a splitting of the signal, assigned to acetic acid for samples 3 and 25 (Fig. 5). That implies the possibility of overlap of the signals of two different compounds, authors should comment on that. 

Author Response

We thank the reviewer for the thoughtful review of our manuscript. We respond to the specific comments below:

  1. In this study we introduce PCA of GCMS data as a means to identify suspect values which can then be investigated by NMR. In this case, we feel that PCA of the NMR data is unnecessary. Wording to this effect has been added to the discussion section (lines 370-372).
  2. The NMR system used does not allow for pulse shaping, therefore multi-signal suppression is not possible. We agree that a multi-point presaturation procedure would be advantageous.
  3. Additional information regarding the NMR experimental conditions has been added to the text (line 128-134).
  4. The suggest change has been made (line 139)
  5. We didn’t do statistical analysis on the NMR data – no binning was performed. Integrations were compared to the integration of the maleic acid internal standard. We followed the classic qNMR approach to determine mole ratios and then mass values are obtained from molecular weights (See section 2.5).
  6. Looking at the other peaks in the spectra of sample 3 and 25 they all show this “splitting” which points to it being a shim issue effecting the peaks across the board. A comment about this has been added to the discussion (line 241 and 242).

Round 2

Reviewer 1 Report

No suggestions

Reviewer 2 Report

I still believe that multiple-solvent suppression is possible on the NMR spectrometer employed by the authors (one may use a standard 1D-NOE sequence to presaturate each signal of both solvents), but I agree that the presented results are convincing enough even in the absence of these data.